# Lack of Association between the Reasons for and Time Spent Doing Physical Activity

**DOI:** 10.3390/ijerph17186777

**Published:** 2020-09-17

**Authors:** Màrius Domínguez-Amorós, Pilar Aparicio-Chueca

**Affiliations:** 1Department of Sociology, Faculty of Economics and Business, University of Barcelona, 08034 Barcelona, Spain; mariusdominguez@ub.edu; 2Department of Business, Faculty of Economics and Business, University of Barcelona, 08034 Barcelona, Spain

**Keywords:** physical activity, active live, motivational factors, health, Europe, structural equation modeling

## Abstract

Low levels of Physical Activity (PA) and sedentarism are associated with the onset of different pathologies and health problems. Regular physical activity has been linked with being beneficial to the health of the general population. Within this framework of analysis, the aim of the present study was to analyze the association between the time spent doing physical activity and the expressed motives for doing so, from which the innovative aspect of the paper emerges: the use of the time spent doing PA as a study variable of the phenomenon. The data analyzed come from the latest special Eurobarometer survey about the sport and physical activity done in Europe. Using an exploratory factorial analysis and a structural equations model, a six-dimensional factorial model was found that explains the reasons for doing PA, demonstrating that there is no relationship between the reasons for and time spent doing PA. The motivation is not a variable that explains the time spent doing PA, and another type of variable must be used to explain the phenomenon if PA is to be incentivized. Weaknesses of the study are that it works with individuals as a group and that the fundamental dependence on age is not introduced, which could determine interest in practicing PA. Similarly, the impact of the conditions of implementing PA, education, and family history should also be introduced into the model.

## 1. Introduction

One of the health goals of all advanced societies is to increase Physical Activity (PA) among the population, with effective policies and strategies designed as much to stimulate PA as to overcome the obstacles to doing it. As the basis for the design of these strategies and policies, different authors have highlighted the need to know the patterns of PA among the population, in addition to the factors and reasons that lead an individual to begin to do PA and to keep it up or give it up [1,2,3,4].

Doing physical activity has become an increasingly important social issue in recent decades because, among other reasons, physical inactivity and sedentary lifestyles have contributed to the growing prevalence of pathologies, such as obesity, diabetes mellitus, heart disease, hypertension, and cerebrovascular events, today the scourge of modern societies [5]. Low levels of PA and sedentarism have been associated with the onset of different pathologies and health problems [6,7]. Numerous studies have demonstrated that doing regular PA is beneficial for body composition [8,9,10], bone health [11,12], psychological health and stress [13,14], and cardiorespiratory capacity, among others [15,16,17,18]. The prevalence of these illnesses is still increasing, leading to greater morbidity and mortality, and the consequent urgent need to implement effective PA programs to reduce the sedentary behavior of the population [19].

In this paper, physical exercise is defined as a physical activity that is done with a specific aim [20]. In other words, the activity is pre-planned and organized to improve or maintain one or more aspects that determine an individual’s physical condition [21,22,23]. Regular exercise is one of the main instruments available to the population to promote and maintain good health. Motivation is considered to be a process that stimulates and directs behavior towards the aim or goal of an activity that is instigated and sustained, while obstacles are understood as barriers or impediments that lead to not doing an activity, and which therefore have a fundamental impact on starting and consolidating PA habits [20,24,25].

Many studies in recent years have uncovered a multitude of motivational factors that directly or indirectly intervene in people’s interest and participation in doing physical exercise [20,21,22,23,24,25]. Regarding the variables that generate motivation, reference [22,23] point to having fun and occupying free time, maintaining physical fitness and appearance, health, and an enjoyment of sports, in that order, as the main reasons why people do physical exercise in their free time.

The study of the reasons for doing PA can be approached from different perspectives. A first group of studies analyze the association between motivational factors and sociographic variables and lifestyles (gender differences, adolescents, university students, retired people, etc.) [26,27,28,29,30]. A second group focus on analyzing the reasons for doing PA in a specific city or country [31,32,33,34,35,36,37]. And a third and last group relate the reason for doing PA with health. In this case, the studies seek to establish relationships between chronic illnesses, such as diabetes, obesity, and cancer, among others, and motivation [38,39,40,41,42,43,44]. The present study wanted to add a new analysis group, relating motivation with the time spent doing PA. This is a well-used line in other areas of social science studies but has so far not been developed in the field of sport.

The study of the time spent doing PA is of interest because it contributes a new variable to the study of PA, even though it is one that is already used in other branches of the social sciences. Within this analysis framework, the general aim of the study was to analyze the existing relationship between the time spent doing PA (either moderate or vigorous) and the expressed reasons for doing so, the importance of which is rooted in improving the analysis of the phenomenon, thus increasing knowledge about PA to be able to effectively promote its performance and improve the health of society.

The main hypothesis of the study was that there are significant differences in the time Europeans dedicate to PA depending on the motivation for doing it. The study specifically wanted to show whether the people who engage in PA for health reasons spend more time doing it, either vigorously or moderately, given that benefitting to health is an intrinsic and substantial reason for doing PA.

## 2. Materials and Methods

The methodology of the study was quantitative, based on the statistical analysis of a social survey, the 2017 Eurobarometer [45] on sport and physical activity, as a secondary information source. The statistical data analysis carried out consisted of (1) the univariate analysis of the variables, calculation of the time measures, and the transformations of the variables to meet the assumptions of linearity, normality, and missing values; (2) the exploratory factor analysis of the dimensional structure of the reasons for performing PA; and (3) the Structural Equation Models analysis with the integrated application of a confirmatory factor analysis and path analysis to validate the explanatory model of the time spent doing PA.

### 2.1. Data

The data analyzed came from the special Eurobarometer on sport and physical activity carried out by the European Union in all 28 of its member states. The last edition was presented in December 2017 [45]. The database was comprised of the information given face-to-face by 28,031 individuals. It was a representative sample of all the European countries, with the individuals selected stating that they had done either moderate or vigorous PA in the previous week, in addition to reporting the time spent doing both. The subsample was comprised of 9404 individuals. The Eurobarometer survey focuses on 7 different areas related to physical activity: frequency, intensity, time spent doing it, place where citizens do it, motives for and obstacles to doing it, voluntary participation in it, and how they perceive the policies implemented by the local government regarding doing physical activity in their local area (Table 1).

### 2.2. Measures

At the methodological level, this study uses two different groups of variables. The first group is comprised of all the variables related to the time Europeans spend doing PA. The second group is made up of the variables related to the reasons for doing PA. Regarding the first group (the dependent variables of the analysis performed), one of the key elements and a topic of debate in the scientific community is the operationalization of the measurement of PA. While some authors state that the most objective way of measuring PA is by monitoring heart rate using accelerometers, pedometers, or indirect calorimetry, others affirm that self-reporting methods or questionnaires are sufficient [46,47,48,49,50].

One of the instruments used to measure regular physical activity is the International Physical Activity Questionnaire (IAPQ) [51,52]. Both methods to operationalize and calculate physical activity levels involve technical difficulties and are highly costly in terms of the fieldwork required to carry out international level comparative studies. Therefore, to the authors knowledge, there are currently no published studies at the European level that use these methodologies due to their high cost. The special Eurobarometer on sport and physical activity is one of the most used tools in other important social surveys of public opinion, collecting data about the frequency (days per week) and minutes per day that Europeans do PA, differentiating between moderate and vigorous PA [53].

The proposed calculation method in this study is the operationalization of the time spent doing PA in minutes per week, multiplying the frequency of doing PA in days per week by the minutes per day spent doing moderate or vigorous physical activity. Regarding the time variable moderate and vigorous PA, logarithmic transformation was applied due to the assumptions of normality and linearity not being met. The possible existence of univariate and multivariate atypical values was also checked by means of standard normal deviations, such as the Mahalanobis distance.

Among the second large group of variables to be analyzed, the independent variables, there were those related to the reasons Europeans give for doing PA. In this case, and as a closed-ended question, those surveyed were asked directly about their most usual reasons for doing sport and PA. This multi-response question was transformed into dummy variables, one for each reason, and the items with a response level below 5% were discarded.

### 2.3. Statistical Analysis

Europeans spend an average of 223.34 min per week doing vigorous PA and 245 min per week doing moderate PA (Table 2).

The most given reason why respondents did PA was to improve their health (58.73%), followed by for fitness (52.26%). Other popular reasons with percentage response rates of between 30 and 40% were to relax, to improve their physical performance, to have fun, and to control their weight. Table 3 shows the descriptive statistics of the reasons given, together with the average time spent doing either vigorous or moderate PA. The results show that the most time spent doing both moderate and vigorous PA is when it is being done to learn new skills and to improve self-esteem.

An exploratory factorial analysis (Varimax rotation) was conducted to uncover the underlying structure of a reasons for doing PA. This factor structure is used in confirmatory factor analysis. Second, Structural Equation Models (SEM) were generated, using the AMOS 25 program (IBM Corporation-AMOS Development Corporation, Meadville, PA, USA). In brief, by means of the integrated application of various methods, such as confirmatory factor analysis and path analysis, the structural equation model is a statistical technique to assess the validity and reliability of theoretical models composed of latent constructs. Figure 1 shows the hypothesized model, consisting of two components: the measurement model, which defines six unobservable latent variables using one or more observed variables (reasons for doing PA); and the structural model, which imputes relationships between latent variables. Modeling the relationships between latent variables enables the estimated effects of the time spent doing PA to not depend on the errors of means, therefore considering the error terms and estimating the effects simultaneously. The structural part of the model considers that the reasons explain the time spent doing moderate or vigorous PA, latent observed variables based on the two-time indicators, considering a relationship between their error terms.

The aims of this study were twofold. First was to validate the measurement model by means of a confirmatory factorial analysis of the reasons for doing PA (the left-hand part of Figure 1) as the independent variables. Second was to test the structural model (the right-hand part of Figure 1) using an explanatory model, selecting the time spent doing moderate and vigorous PA as the dependent variables. The estimation method used was the Weighted Least Squares (WLS) [54], a method that appears in the AMOS 25 program [55] under the label Asymptotic Free Distribution method (ADF) [56]. Among other advantages, this method allows dichotomous variables and continuous variables that do not fit with the criteria of normality to be introduced into the analysis. It is one of the methods most used and recommended when the data do not meet the assumption of normality.

## 3. Results

### 3.1. Exploratory Factorial Analysis with Varimax Rotation (Principal Component Analysis)

The Exploratory Factorial Analysis (EFA) estimated the factorial structure of the reasons for doing AP in six factors or dimensions. The data corresponding to the Kaiser-Meyer-Olkin index (KMO), at 0.80, and the Barlett test with a signification of 0.000, indicated that the factorial solution explaining 70% of the initial variance was adequate. The factorial loads presented in Table 4 account for the structure of the rotated component matrix, showing 6 dimensions of reasons for doing moderate or vigorous PA. The first dimension contained the reasons related to fitness, health improvement, and physical performance; the second dimension included the reasons related to physical appearance; and in the third dimension, there were doing PA for fun and for the social aspect. The reasons to increase skills and competence, to counteract ageing, and to relax shaped dimensions was 4, 5, and 6, respectively. Notably, the reason to increase self-esteem was included in two of the dimensions: in dimension 2, together with the reasons to do with physical appearance, and in dimension 4 regarding increasing skills and competence.

### 3.2. Structural Equation Modeling (SEM)

#### 3.2.1. Measurement Model: Confirmatory Factor Analysis (CFA)

Based on the results of the initial exploratory factorial analysis, the six-factor model found was tested to see if it fitted the data and if the two models had an identical structure or whether significant differences between moderate PA and vigorous PA could be identified. The second question was whether there was a significant covariance between these factors, or if they were similar in the two models. To meet this objective, two measurement models were developed, one for the time spent doing vigorous PA and the other for the time spent doing moderate PA. The results of the two models showed that they have identical structures, so we considered it appropriate to present a single measurement model that took the population’s reasons for doing both moderate and physical activity into account. The measurement model fitted the 6 latent factors, with a Chi-square of 348,944, 31 degrees of freedom, and a p-value of less than 0.001.

Figure 2 shows the standardized regression coefficients, the correlations among the exogenous variables, and the estimates of squared multiple correlations.

The results obtained enabled us to validate the factorial structure of the measurement model based on the six latent factors identified in the previous phase. The standardized loadings for most of the items are significantly close to 0.5, except for the reason new skills, which was also the reason given with the lowest explained variance percentage. In addition, worthy of note was the high correlation between the latent factors F6 (fitness and health improvement) and F2 (physical appearance), F5 (self-esteem and new skills), and F2 (physical appearance), all of which were higher than 0.50.

#### 3.2.2. Structural Model

Figure 3 and Table 5 show the results of the structure model. The explanatory capacity of the reasons according to the time spent doing vigorous and moderate PA, using their latent variables, can be observed.

The model’s global fit showed that the hypothesized model fitted significantly. Moreover, the analysis of the modification indexes showed that no re-specification of the model or substantial changes in the estimated parameters could be made. The results show that the explanatory capacity of the reasons for doing PA according to the time spent doing it was low, at approximately 2% for both types of PA. Furthermore, the estimate of the parameters of the model showed that the correlation between the error of the endogenous variables (which does not explain the time spent doing the two types of PA) was correlated (0.53), indicating that what was not explained by the reasons for spending time doing PA had a shared identity with both vigorous and moderate physical activity.

It is noteworthy that the latent variables that had a significant effect on the time spent doing PA. For both moderate and vigorous PA, the latent factors of physical appearance and self-esteem had a significant effect on the time spent per week doing PA. Contrarily, however, while the reason physical appearance implied a reduction in the time spent doing PA (−0.831 and −0.544 for moderate and vigorous PA, respectively), the reasons self-esteem and new skills implied an increase in the time spent doing PA (0.591 and 0.649 for moderate and vigorous PA, respectively). Furthermore, in the case of vigorous PA, the reason to relax also had the effect of reducing the time spent doing PA, although less so than physical appearance (−0.078).

## 4. Discussion

This study aimed to open up a new perspective, the study of time and motivation, and the results contribute to the literature with this new variable to consider within the study of PA. Specifically, the time variable in studies alongside the other three perspectives. An important limitation of the study is the lack of temporary data (at different points in time) on the spent in PA that allows to analyze the time before and after the indicated reasons appear and the previous physical experience. For example, in case of health problems: for people who indicate health reasons as a reason to perform PA, are there differences in time when compared to the time spent before having health problems?

The first objective set in this study was to develop a factorial structure that reduced the number of reasons why Europeans do PA. The results showed a robust factorial structure made up of 6 dimensions: fitness and health, appearance, fun and friend (F&F), self-esteem, aging, and relaxation. This dimensional structure coincided with the motivational structures studied in the bibliography review [22], allowing us to create a factorial model that summarized the reasons for doing PA.

The second objective was to determine if there were two different model structures for vigorous PA and moderate PA, and if the model was sound, or in other words, to identify if the motivational structure was different for people doing vigorous PA and moderate PA. As the confirmatory analysis showed, there were not two different structures and the two types of activity were explained by the same measurement model. The measurement model was robust and explained this behavior (the correlation between the times implied that both vigorous and moderate exercise were explained by the same model).

Dimension F2, physical appearance, was highly correlated with the dimension of health and exercise. Therefore, the Europeans that stated that they do PA for their appearance and to indirectly acquire new skills were also doing it for health reasons. The model also showed another correlation between the factors of appearance (F2) and self-esteem (F5). This result was to be expected since people concerned with their appearance and who do PA to improve it are also indirectly improving their self-esteem. These correlations are important given that they can be used to re-focus policies to incentivize doing PA to emphasize not only the health aspect but also the factors of appearance and self-esteem.

The third and last objective was to analyze whether the reasons for doing PA explained the time spent doing it, either vigorously or moderately. Previous literature suggests a positive relationship exists between PA and health. This study hypothesized that people who are motivated by health to engage in PA, would be significantly more time doing PA. However, this hypothesis was rejected given no such relationship was demonstrated in this sample. The lack of more information about the reasons. For example, in the case of the health reason, the question here is why are people ‘motivated’ or not by health: motivated by health in the context of family illness, motivated in health as prevention or as treatment, or have a health condition which restricts their ‘time doing’ PA.

The last structural equations model indicated that there was no relationship between them, or, in other words, that the analysis of the reasons for doing PA did not involve studying the time spent doing it. This finding suggests that to analyze the time spent doing PA, it would be more appropriate to relate it to other types of variables, such as demographic ones or types of diseases, among others. Incorporating the time spent doing PA into studies on PA is also a new line of investigation that can help to know the phenomenon better and can better incentivize the public policies that promote doing PA, thus improving the health of the population. The time spent doing PA and the times of day must be known to create both public and private spaces that help to promote PA and, indirectly, people’s health, social integration, and self-esteem.

The main limitation of the study is the way the survey was carried out, to the effect that when asked their reasons for doing PA respondents were not asked to differentiate between vigorous and moderate PA. Therefore, their self-reported reasons were for doing PA in general. Other limitations stem from the construction of the variable “time spent” and from the subsample chosen, which was made up of Europeans that do PA and who had done vigorous or moderate PA in the previous seven days.

As a future line of research, it would be appropriate to analyze the motivation of health on inactivity and sedentary behavior, as well as participants’ beliefs and perceptions enabling and prohibiting PA engagement (for example intrinsic/extrinsic motivation).

## 5. Conclusions

The findings are important because they explain the time spent doing PA and the reasons for doing so irrespective of age, gender, class variables, lifestyle, health, and ethnicity, among other factors, using a European database. Despite the limitations, the results contribute to the literature by providing a new variable to consider within the study of PA. The main novelty of the study is that it introduces the variable time spent doing PA, analyzing the relationship between this variable and the reasons for doing PA. It seems logical to think that people who do PA for health reasons spend more time doing it. However, the results of the study contradict this main study hypothesis. The motivation is not a variable that explains the time spent doing PA and another type of variable must be used to explain the phenomenon if PA is to be incentivized. There is no relationship between the reasons for doing PA and the time spent doing it. Weaknesses of the study are that it works with individuals as a group and that the fundamental dependence on age is not introduced, which significantly determines interest in implementing PA. Similarly, the impact of the conditions of implementing PA, education, and family history of PA should also be introduced into the model.

As stated in the introduction to this paper, analysis of the reasons for doing PA has been previously carried out from three different perspectives. This study aimed to open up a new perspective, the study of time and motivation, and the results contribute to the literature with this new variable to consider within the study of PA. It seems logical to think that people who do PA for health reasons spend more time doing it. However, the results of the study contradict this main study hypothesis, demonstrating that there is no relationship between the reasons for doing PA and the time spent doing it. Nonetheless, the conclusion drawn is that it is important to introduce the time variable in studies alongside the other three perspectives.

## Figures and Tables

**Figure 1 ijerph-17-06777-f001:**
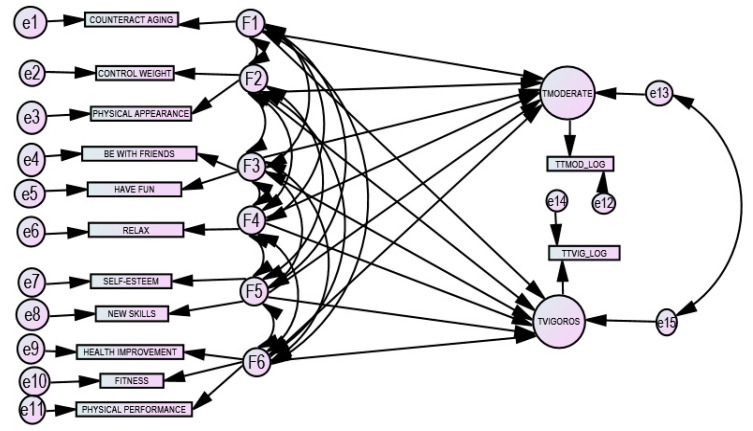
Hypothesized model: measurement model and structural model F1 to F6: latent variables; e1 to e11: errors in variables.

**Figure 2 ijerph-17-06777-f002:**
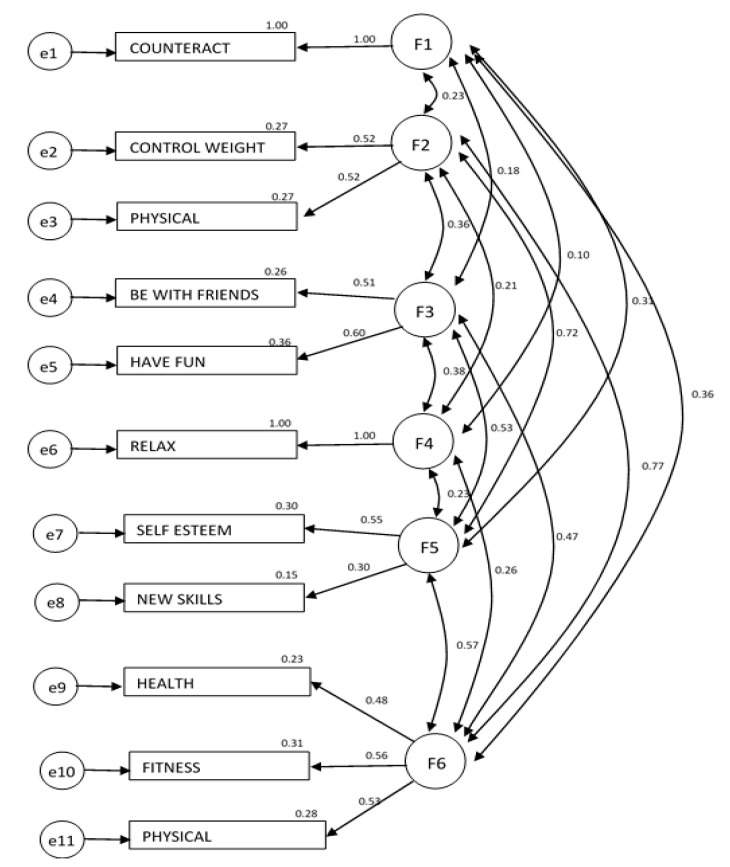
Results of the measurement model. Confirmatory factor analysis. Chi-square = 348.944; Probability Level = 0.000; CFI = 0.912; RMSEA = 0.0033. The standardized regression coefficients and correlations among the exogenous variables are shown next to the arrows. The estimates of the squared multiple correlations are reported below the construct names. The ovals represent the latent variables and the rectangles represent the observed variables. All the regression weights are significantly different from zero at the 0.001 level. Loading is fixed at the value of 1 in the non-standardized solution.

**Figure 3 ijerph-17-06777-f003:**
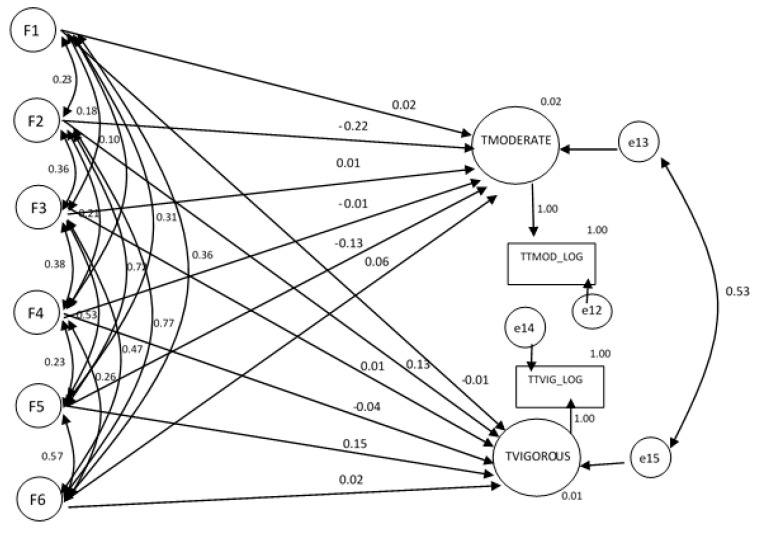
Results of the structural model. Standardized weights. CHI = 397.260; Probability Level = 0.000; CFI = 0.942; RMSEA = 0.0030. The standardized regression coefficients are shown next to the arrows. The estimates of squared multiple correlations are reported below the construct names. The ovals represent the latent variables and the rectangles represent the observed variables. Loading fixed at the value of 1 in the non-standardized solution.

**Table 1 ijerph-17-06777-t001:** Socio-demographic characteristics of the subsample that does moderate and vigorous Physical Activity (PA).

Variables	Percent
Gender	
Men	52.5
Women	47.5
Age (Mean)	46.75
15–24 years	11.9
25–39 years	25.7
40–54 years	26.2
55 years and older	36.2
Marital status	
Unmarried	20.4
(Re-)Married/Single with partner	66.3
Divorced or separated	6.9
Widowed	5.4
Other	0.9
Refusal	0.2
Age education	
Up to 15 years	8.1
16–19	40.9
20 years and older	40.8
Still studying	8.8
No full-time education	0.3
Refusal	0.1
Respondent occupation scale	
Self-employed	9.3
Managers	14.6
Other white-collar workers	13.3
Manual workers	22.8
House persons	3.1
Unemployed	5.2
Retired	22.9
Students	8.8
Type of community	
Rural area or village	30.0
Small/medium-sized town	42.5
Large town	27.4
Social class—Self-assessment	
Working class	19.6
Lower middle class	12.8
Middle class	52.4
Upper middle class	11.3
Higher class	1.1
Other	0.1
None	0.8
Refusal	0.5

**Table 2 ijerph-17-06777-t002:** Statistics of the time spent doing PA (minutes per week). Subsample of individuals that do vigorous and moderate PA.

Statistics of the Time	Vigorous PA	Moderate PA
Average number of minutes of PA/week	223.34	244.84
Standard deviation	205.16	216.34
Average number of minutes of PA/day	64.82	61.26

**Table 3 ijerph-17-06777-t003:** Statistics for the time spent doing vigorous or moderate PA (minutes per week) according to the reason for doing it. Subsample of subjects that did vigorous or moderate PA.

Reason for Doing PA	Percentage	Time Spent Doing PA Vigorous	Tiene Doing PA Moderate
Improve health	58.73	216.60	239.23
Physical appearance	23.35	216.87	231.16
Counteract aging	21.16	212.71	241.30
Have fun	33.17	223.09	248.36
Relax	39.63	213.17	241.24
Be with friends	22.89	218.01	239.77
Physical performance	33.41	223.86	242.97
Fitness	52.26	210.04	234.28
Control weight	30.42	202.28	224.09
Self-esteem	16.49	229.22	248.77
New skills	9.70	249.04	257.46

**Table 4 ijerph-17-06777-t004:** Rotated components matrix of the Exploratory Factorial Analysis (EFA). Factors, percentage of explained variance, and factorial loads. Subsample of individuals that did moderate or vigorous PA.

Motivations or Reasons Given for Doing PA	Factorial Loads Percentage of Explained Variance					
	1	2	3	4	5	6
	13.87%	12.21%	12.05%	10.90%	9.28%	9.20%
Fitness	0.719					
Health improvement	0.696					0.216
Physical performance	0.642			0.287		
Physical appearance		0.754		0.222		
Control weight		0.722			0.226	
Be with friends			0.832			
Have fun			0.728			0.233
New skills				0.831		
Self-esteem		0.427		0.580		
Counteract aging					0.943	
Relax						0.928

Extraction method: principle component analysis. Rotation method: Varimax with Kaiser normalization.

**Table 5 ijerph-17-06777-t005:** Regression weights and standardized estimate with the *p*-values of the factors and the latent variables of the time spent doing PA.

Observed Variables	Effect	Latent Variables	Estimate	Standardized Estimate	*p*
Moderate time	←	F1	0.04	0.02	0.136
Moderate time	←	F2	−0.83	−0.22	***
Moderate time	←	F3	0.04	0.01	0.760
Moderate time	←	F4	−0.02	−0.01	0.401
Moderate time	←	F5	0.59	0.13	**
Moderate time	←	F6	0.25	0.06	0.116
Vigorous time	←	F1	−0.02	−0.01	0.510
Vigorous time	←	F2	−0.54	−0.14	**
Vigorous time	←	F3	0.05	0.01	0.682
Vigorous time	←	F4	−0.08	−0.04	***
Vigorous time	←	F5	0.65	0.14	***
Vigorous time	←	F6	0.07	0.02	0.639

** *p* < 0.01. *** *p* < 0.001.

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
