# Peer review of "Lack of Association between the Reasons for and Time Spent Doing Physical Activity"

_ijerph, 2020, doi:10.3390/ijerph17186777_

Round 1

Reviewer 1 Report

Thank you for your article submission, I hope you find the following commentary constructive.

Title- is verbose. Please simplify.

Please ensure all abbreviations have a capital letter one first use, e.g.

physical activity (PA) should be Physical Activity (PA), check throughout manuscript.

Please ensure paper is written in past tense throughout

Please do not refer to first person, or we.

Abstract would benefit from further structure and more detail please.

Introduction. You make some bold claims, and must back up with citations. For example the opening sentence “

One of the health goals of all advanced societies is to increase PA among the population (CITATION FOR ALL ADVANCED SOCIETIES?), with effective policies and strategies (WHICH) designed as much to stimulate PA as to overcome the obstacles that impede it.” WHAT OBSTACLES

Please be consistent in use of abbreviations

“ of different pathologies and health problems “ such as ?

“ The prevalence of these illnesses is still increasing” which illnesses?

In this study, not in this paper

“Many studies in recent years have uncovered a multitude of motivational factors that directly or indirectly intervene in people’s interest and participation in physical exercise.” Citations please to these many studies

Please state clear the study aims, please state the study hypothesis

Please clearly state the study design.

Please consider clarifying

“Both methods to operationalise and calculate physical activity levels involve technical difficulties and are highly costly in terms of the fieldwork required to carry out international level comparative studies, and this is not their specific objective.    THIS IS NOT? what is not?

Therefore, there are no European-level studies that use these methodologies.” I don’t agree? Please clarify what is unique here?

The special Eurobarometer on sport and physical activity is still the tool most used in other major public opinion social surveys. Citations please

Please clarify the measures actually used and stat their validity and reliability with reference to cited evidence.

You have presented initial results within the methodology section, please clean up the methodology section and present as per journal standard.

Please ensure your methods are cited so clear to the reader what you did e.g generalised least squares (GLS).

Please confirm when this data was collected and analysed.

Line 296 use of ‘and so on’, example your writing style needs to be more academic and less verbose throughout.

Please relate the discussion more clearly to hypothesis and how this study adds value to the academic evidence base.

What are the implications for real-world PA?

What is the relevance for future research?

Please expand the study limitations.

Author Response

Dear Review,

Please find enclosed a revised version of the article, coded:  IJERPH-900946, entitled “Association between the reasons for and time spent doing physical activity, two variables with little relationship. An approach from the Eurobarometer Survey”, together with our response to your comments.

We are very grateful to the reviewers, whose comments have helped to improve the article. You will see that this manuscript has improved substantially. Please, we want to thank you for the time and enthusiasm you have dedicated to doing the review. The following pages describe our detailed responses to each of your comments, which have also been changed in the manuscript.

Yours sincerely,

The authors

RESPONSE TO REVIEWER’S #1 COMMENTS

Comments to the Author

Thank you for your article submission, I hope you find the following commentary constructive.

We thank the reviewer for this positive comment.

Title- is verbose. Please simplify.

We would like to thank the reviewer for this comment. We have valued your comment and reflected on it and the title of the article has been simplified: “Lack of association between the reasons for and time spent doing physical activity”.

Please ensure all abbreviations have a capital letter one first use, e.g.physical activity (PA) should be Physical Activity (PA), check throughout manuscript.

Thank you very much for the clarification. All text has been reviewed and your comment has been followed.

Please ensure paper is written in past tense throughout. Please do not refer to first person, or we.

We appreciate your comments and the text has been changed in the past and in the impersonal person. Moreover, a translator has revised the text.

Abstract, would benefit from further structure and more detail please.

We would like to thank the reviewer for this comment. The abstract was not made longer so as not to exceed the maximum number of words. Your comment has been taken into account and more information has been added.

Introduction

  • You make some bold claims and must back up with citations. For example, the opening sentence “One of the health goals of all advanced societies is to increase PA among the population with effective policies and strategies designed as much to stimulate PA as to overcome the obstacles that impede it.”

The authors want to thank you for your hard reading of the work and thank you for all the comments in this section. Following his suggestions, some statements have been expanded and specified and bibliographic references added. Such suggestions increase the quality of study.

  • Please be consistent in use of abbreviations

As mentioned in a previous comment, all the abbreviations used have been revised.

  • of different pathologies and health problems, such as ? The prevalence of these illnesses is still increasing” which illnesses?

We would like to thank you for your comment. The bibliographic references 8 to 18, refer to the types of diseases and in the text it is explained in lines 33-40.

  • In this study, not in this paper

Thank you very much. Paper has been changed to study. [Line 12, 14, 20, 56, 62, 72, 105, 121, 277, 318, 322]

  • “Many studies in recent years have uncovered a multitude of motivational factors that directly or indirectly intervene in people’s interest and participation in physical exercise.” Citations please to these many studies

Thanking you for your comment, bibliographic citations on this topic have been added on line 50.

  • Please state clear the study aims, please state the study hypothesis

Thank you very much for the comment. It is a beginner's mistake. A general hypothesis has been drawn up and introduced at the end of the Introduction section. In the conclusions, this hypothesis has been related to the results obtained.

Materials and Methods

  • Please clearly state the study design. Please clarify the measures actually used and stat their validity and reliability with reference to cited evidence.

Thank you very much for your comment. A new text has been added at the beginning of section 2. There the study design is better explained. [Line 78-85]

  • Please consider clarifying: “Both methods to operationalise and calculate physical activity levels involve technical difficulties and are highly costly in terms of the fieldwork required to carry out international level comparative studies, and this is not their specific objective.    THIS IS NOT? what is not?

We appreciate your comment and the sentence has been rewritten for better understanding. Line [114-116]

  • Therefore, there are no European-level studies that use these methodologies.” I don’t agree? Please clarify what is unique here?

Thank you very much for your appreciation. In this case we believe that it is a problem of translation into English. What was meant is that given the high cost of the aforementioned methodologies, there are no comparative studies at European level with these methodologies. The sentence has been worded better. Line [116-117]

  • The special Eurobarometer on sport and physical activity is still the tool most used in other major public opinion social surveys.

Thank you very much for your appreciation. The statement has been toned down. [Line 117 and 118]

  • You have presented initial results within the methodology section, please clean up the methodology section and present as per journal standard.

Thank you very much for your comment. This information has been transferred to the Statistical Analysis section. [Line 134-150]

  • Please ensure your methods are cited so clear to the reader what you did e.g generalised least squares (GLS).

An translate error has been detected in identifying the estimation method used. It has been corrected and detailed clearer for readers. [Line 174]

  • Please confirm when this data was collected and analysed.

Thank you very much for the comment. The data was collected in December 2017 [line 90-91] and the study was analysed in spring 2020.

  • Line 296 use of ‘and so on’, example your writing style needs to be more academic and less verbose throughout.

Thank you very much for the advice. The text has been revised again and adapted to a more academic style. 

Discussion and Conclusions

  • Please relate the discussion more clearly to hypothesis and how this study adds value to the academic evidence base. What are the implications for real-world PA? What is the relevance for future research?

Thank you very much for these questions, which have helped us to rethink about the Discussion section. In the Conclusions section, a text has been added that relates the main hypothesis of the study with the results. Likewise, a paragraph has been added on the real and social implications of the study and the possible future lines of research. [Lines 273-313]

Please expand the study limitations.

We want to thank you for the comment. Study limitations have been expanded. [Line 318-328]

Reviewer 2 Report

Although there is no doubt about the influence of regular realization of physical activities on the current state of a person, and the majority of the population is aware of this fact, the realized movement regime is still defi deficit. Therefore, studies trying to identify the causes are up-to-date and necessary. Unfortunately, in most cases they end up as descriptive, they state the status quo and very rarely offer a workable solution. The same is true of the study under investigation, which identifies, on the basis of a statistical model, three clusters of causes resulting in the active implementation of PA. Another weakness I see is that it works with individuals as one group and does not even in the discussion try to mention the fundamental age dependence, which significantly determines the interest in the implementation of PA. Similarly, it is necessary to at least discuss the impact of conditions on the implementation of PA, education and family history of movement. Formally, it is necessary to specify the abstract. I recommend formulate working hypotheses at the end of the introduction and answer these in the conclusion.

Author Response

Dear Review,

Please find enclosed a revised version of the article, coded:  IJERPH-900946, entitled “Association between the reasons for and time spent doing physical activity, two variables with little relationship. An approach from the Eurobarometer Survey”, together with our response to your comments.

We are very grateful to the reviewers, whose comments have helped to improve the article. You will see that this manuscript has improved substantially. Please, we want to thank you for the time and enthusiasm you have dedicated to doing the review. The following pages describe our detailed responses to each of your comments, which have also been changed in the manuscript.

Yours sincerely,

The authors

RESPONSE TO REVIEWER’S #2 COMMENTS

Although there is no doubt about the influence of regular realization of physical activities on the current state of a person, and the majority of the population is aware of this fact, the realized movement regime is still defi deficit. Therefore, studies trying to identify the causes are up-to-date and necessary. Unfortunately, in most cases they end up as descriptive, they state the status quo and very rarely offer a workable solution. The same is true of the study under investigation, which identifies, on the basis of a statistical model, three clusters of causes resulting in the active implementation of PA.

I fully agree with the reviewer on these statements. First, the undisputed importance of regular PA performance in the health of the population. Secondly, that this importance implies that it is a topic research of actuality and thirdly, that these studies are fundamentally descriptive and that they do not provide direct solutions applicable to society. However, we consider that these studies make new contributions that will allow, in the future, to advance to achieve practical solutions.

Another weakness I see is that it works with individuals as one group and does not even in the discussion try to mention the fundamental age dependence, which significantly determines the interest in the implementation of PA. Similarly, it is necessary to at least discuss the impact of conditions on the implementation of PA, education and family history of movement. Formally, it is necessary to specify the abstract.

I fully agree with these limitations. Following the reviewer's comment, it has been incorporated into the text in the conclusions section and also in the abstract.

I recommend formulate working hypotheses at the end of the introduction and answer these in the conclusion.

We fully agree with these limitations. Following the reviewer's comment, its have been incorporated into the text in the conclusions section and also in the abstract..

Reviewer 3 Report

This is an interesting manuscript with a novel approach to considering reasons for doing PA. Overall, the manuscript is informative. In its current form the two issues should be corrected by the authors. Firstly, the methods section is confusing and not clearly laid out. Methods section also mixes results, which should be included in the results section. There is no mention about participants providing informed consent. This should be added. Secondly, the discussion section lacked reflection of the current results against the previous literature, especially of those included in the introduction section. This should be added in the discussion. For more detailed comments please see under.

Heading (title)

Please could you reconsider the wording – currently this is confusing. Heading mentions “reasons” i.e. plural – and PA time but describes these as two variables – however, this is actually more than just two variables.

Abstract

Line 20: “Therefore…” this conclusion is somewhat difficult to follow. Previously (line 15) authors mention expressed motives – do the authors mean here motivation in general or the different expressed motivations?

Introduction

Line 29: “…impede it.” Please add reference – not only at the end of the paragraph.

Line 35: “…nowadays the scourge of modern societies [5]” As it is currently written it appears that the authors refer especially the cerebrovascular incidents as the scourge of modern societies. Please check the wording if all the illnesses are meant as scourges – and please add a comma at the end of the sentence.

Line 40: “…and the consequent the urgent need…” please consider rewording – it reads as part of the sentence would be missing.

Line 45: “Motivation is considered to be a process that “stimulates and directs behaviour towards…”. Please could you consider rewording the sentence as it is very difficult to follow. The definition provided in here is difficult to follow as it mixes motivation and barriers to physical activity (but leaves out enablers). Also, is this the original or translated quote?

Line 55: Please could the authors consider rephrasing - It is unclear whether the 3 categories mentioned are devised by the authors or are already mentioned in the literature – and why do the authors want to add a fourth category.

Please could you provide not only general study aim but also specific study aims as well as any hypothesis tested.

Methods

Please could you provide more information about Eurobarometer survey – how the data was collected, how participants were contacted, informed consent etc. It is also rather unclear what is the difference between the full- and sub-samples and which sample was used the current study. The study questionnaire (or rather the individual questions) used for data collection should be described in more details. Were any composite constructs formed from the individual questions?

Table 1 and 2 belong to results – not methods – section. Authors should in general ensure that methods sections describe the study methods and not include any results in here (e.g. Line 109, 124). Authors should also consider reorganising the text so that statistical methods are not mixed in descriptions of study methods and materials.

Please could the authors define moderate and vigorous PA as used in this study.

Line 137: Was SPSS used in the Factor Analyses?

Regarding statistical methods – AMOS – requires full dataset – how did the authors deal with any missing data? Or were only participants with full datasets included?

Please could the authors clarify – line 161 and methods selected to accommodate data non-normality. Previously it was mentioned that data transformations were performed to improve data normality.

Results

How did the authors solve the issue of one variable loading in two factors?

Line 170: AP?

Line 171: KMO – has this abbreviation been already defined?

Line 214: Health improvement isn’t above 0.5 either

Discussion

Line 258: F&F? Has this been already defined?

The discussion lacks any reflection of the results in lights of the previous literature, included in the introductions. This should be added. In addition, the discussion should reflect how the study adds in the literature and, considering the emphasis in the introduction, their applicability in prevention of diseases associated with lack of PA.

Author Response

Dear Review,

Please find enclosed a revised version of the article, coded:  IJERPH-900946, entitled “Association between the reasons for and time spent doing physical activity, two variables with little relationship. An approach from the Eurobarometer Survey”, together with our response to your comments.

We are very grateful to the reviewers, whose comments have helped to improve the article. You will see that this manuscript has improved substantially. Please, we want to thank you for the time and enthusiasm you have dedicated to doing the review. The following pages describe our detailed responses to each of your comments, which have also been changed in the manuscript.

Yours sincerely,

The authors

RESPONSE TO REVIEWER’S #3 COMMENTS

This is an interesting manuscript with a novel approach to considering reasons for doing PA.

We thank the reviewer for this positive comment.

Overall, the manuscript is informative. In its current form the two issues should be corrected by the authors.  Firstly, the methods section is confusing and not clearly laid out. Methods section also mixes results, which should be included in the results section. There is no mention about participants providing informed consent. This should be added. Secondly, the discussion section lacked reflection of the current results against the previous literature, especially of those included in the introduction section. This should be added in the discussion. For more detailed comments please see under.

We appreciate all your comments to increase the quality of the study and improve it substantially. First, at the beginning of the methodology section a new paragraph has been included that explains the methods used in a clearer and more synthetic way [Lines 78-85]. Secondly - as you say - there were results in the methodology section. To solve this, table 2 and 3 have been moved to the results section, in the same way as the accompanying text [Lines 134- 150]. Third, it is not mentioned that the participants give their informed consent to participate in the study, since it is a European database whose methodology already indicates this situation. But we include a reference about it in the text [Reference 45]. Finally, the discussion has been written reflecting the relationships between the results obtained with the previous literature presented in the introduction [Lines 273-276].

Heading (title)

  • Please could you reconsider the wording – currently this is confusing. Heading mentions “reasons” i.e. plural – and PA time but describes these as two variables – however, this is actually more than just two variables.

We would like to thank the reviewer for this comment. We have valued your comment and reflected on it and the title of the article has been simplified: “Lack of association between the reasons for and time spent doing physical activity”.

Abstract

  • Line 20: “Therefore…” this conclusion is somewhat difficult to follow. Previously (line 15) authors mention expressed motives – do the authors mean here motivation in general or the different expressed motivations?

Thank you very much for the comment. The term "Therefore" has been removed, making the phrase clearer. The authors refer to the different motivations expressed above (line 15) not to motivation in general.

Introduction

  • Line 29: “…impede it.” Please add reference – not only at the end of the paragraph.

We appreciate the comment. The wording of the sentence has been changed [Line 30].

  • Line 35: “…nowadays the scourge of modern societies [5]” As it is currently written it appears that the authors refer especially the cerebrovascular incidents as the scourge of modern societies. Please check the wording if all the illnesses are meant as scourges – and please add a comma at the end of the sentence.

We appreciate your comment. The bibliography in this regard has been revised. To finish the sentence, a period has been added.

  • Line 40: “…and the consequent the urgent need…” please consider rewording – it reads as part of the sentence would be missing.

Thank you very much for the comment. All the text has been sent to a native translator to review all the text.

  • Line 45: “Motivation is considered to be a process that “stimulates and directs behaviour towards…”. Please could you consider rewording the sentence as it is very difficult to follow. The definition provided in here is difficult to follow as it mixes motivation and barriers to physical activity (but leaves out enablers). Also, is this the original or translated quote?

Thank you very much for the comment. As mentioned before, the entire text has been revised again. In addition, the quotes have been removed since it is not an original quote.

  • Line 55: Please could the authors consider rephrasing - It is unclear whether the 3 categories mentioned are devised by the authors or are already mentioned in the literature – and why do the authors want to add a fourth category.

In response to your comment, the three categories mentioned have been designed by other authors and are mentioned in the literature. This study wants to add a fourth category [Lines 53-64].

  • Please could you provide not only general study aim but also specific study aims as well as any hypothesis tested.

Thank you very much for the comment. It is a beginner's mistake. A general hypothesis has been drawn up and introduced at the end of the Introduction section. In the conclusions, this hypothesis has been related to the results obtained [Lines 72-76].

Methods

  • Please could you provide more information about Eurobarometer survey – how the data was collected, how participants were contacted, informed consent etc. It is also rather unclear what is the difference between the full- and sub-samples and which sample was used the current study. The study questionnaire (or rather the individual questions) used for data collection should be described in more details. Were any composite constructs formed from the individual questions?

The authors have considered that such information was not essential in the study, but reference 45 contains all the information it indicates. Based on your suggestion, some of the information has been added to the text [Line 90].

You can find this information in: https://ec.europa.eu/commfrontoffice/publicopinion/index.cfm/survey/getsurveydetail/instruments/special/surveyky/2164

This survey was carried out by TNS Political & Social network in the 28 EU Member States between 2 and 11 December 2017. Some 28,031 EU citizens from different social and demographic categories were interviewed face-to-face at home and in their native language, on behalf of the Directorate-General for Education, Youth, Sport and Culture. The methodology used is that of the Standard Eurobarometer surveys carried out by the Directorate-General for Communication (“Media Monitoring, Media Analysis and Eurobarometer” Unit). It is the same for all countries and territories covered in the survey. A technical note concerning the interviews conducted by the member institutes of the TNS Political & Social network is annexed to this report.

  • Table 1 and 2 belong to results – not methods – section. Authors should in general ensure that methods sections describe the study methods and not include any results in here (e.g. Line 109, 124). Authors should also consider reorganising the text so that statistical methods are not mixed in descriptions of study methods and materials.

Thank you very much for the comment. Agree with the other reviewers. The text has been rearranged, following your comments [Line 134-150].

  • Please could the authors define moderate and vigorous PA as used in this study.

The use of the vigorous and moderate PA classification is commonly used, both in the European Union and in the US. The definition for light intensity activity is an activity that is classified as < 3 METS. One MET, or metabolic equivalent, is the amount of oxygen consumed while sitting at rest. Thus, an activity classified as 2 METS would be equal to 2 times the amount of oxygen consumed while sitting at rest (1 MET). METS are a convenient and standard method for describing absolute intensity of physical activities.

Moderate PA are defined as activities ranging between 3 - < 6 METS. These activities require more oxygen consumption that light activities.  Vigorous intensity activities are defined as activities ≥ 6 METS. Vigorous activities require the highest amount of oxygen consumption to complete the activity.

  • Line 137: Was SPSS used in the Factor Analyses?

Yes, we used IBM SPSS Statistics 26.0

  • Regarding statistical methods – AMOS – requires full dataset – how did the authors deal with any missing data? Or were only participants with full datasets included?

Yes, it is. Only participants with full datasets are analysed.

  • Please could the authors clarify – line 161 and methods selected to accommodate data non-normality. Previously it was mentioned that data transformations were performed to improve data normality.

Thank you very much for your comment. A new text has been added at the beginning of section 2. There the study design is better explained [Lines 78-85].

Results

  • How did the authors solve the issue of one variable loading in two factors?

No variable loads significantly on two factors. Each variable is significantly projected on a single factor (see table 4)

  • Line 170: AP?

It's a mistake. We mean PA. Text has been changed.

  • Line 171: KMO – has this abbreviation been already defined?

Thank you very much. We are sorry for the mistake. The text has been changed to: ”[…] the Kaiser-Meyer-Olkin (KMO) statistic[…]” [Lines 189-190].

  • Line 214: Health improvement isn’t above 0.5 either

You are absolutely right, but 0.48 is very close to 0.5. Taking this comment into account, the wording of the sentence has been changed to the following: "The standardised loadings for most of the items are significantly close to 0.5, except for the reason new skills […]”

Discussion

  • Line 258: F&F? Has this been already defined?

We are very sorry to have put the abbreviation directly. It was an unforgivable mistake. It has been changed in the text to its literal text (fun and friends) [Line 279].

  • The discussion lacks any reflection of the results in lights of the previous literature, included in the introductions. This should be added. In addition, the discussion should reflect how the study adds in the literature and, considering the emphasis in the introduction, their applicability in prevention of diseases associated with lack of PA.

Thank you very much for the comment. You are absolutely right. Text has been added to the discussion section, improving its quality [Lines 273- 313].

Round 2

Reviewer 1 Report

*Please note your review is female not male, her not his. *

"hypothesis of the study is that there are signifi... " use past tense throughout, was not is  (e.g. line 72, line 78)

line 116: amend sentence as follows please: Therefore, to the authors knowledge, there are currently no published studies at the European level tha...

line 153 "First, we carried out an explor..."  consider academic style refer reference to we.  e.g.    An exploratory factorial analysis was conducted...  or simply exploratory factorial analysis demonstrated/ revealed/ conducted .....

please provide reference for Generalised Least Squares (GLS) (REF). and...
The estimation method used was the Weighted Least Squares (WLS), a method that appears in the AMOS 25 program (REF) under the label Asymptotic Free Distribution method (ADF

line 264 "Of particular note are the latent variable" amend ... it is noteworthy that the latent.... 

discussion

"The bibliographic references analysed showed the positive relationship between PA and health.  In other words, the people whose reason for performing PA was health should spend more time doing PA. However, the constructed model shows that this is not the case and that there is no such relationship. "

Are you suggesting here that 

Previous literature (refs from intro) suggests a positive relationship exists between PA and health.  This study hypothesised that people who are motivated by health to engage in PA, would be significantly more time doing PA.  However, this hypothesis was rejected given no such relationship was demonstrated in this sample. 

I would really like you to expand the discussion and bring in interpretation through health behaviour theory.  For example consider the following " out from three different perspectives. This study aimed to open up a new perspective, the study of time and motivation, and the results contribute to the literature with this new variable to consider within the study of PA. It seems logical to think that people who do PA for health reasons spend more time doing it. However, the results of the study contradict this main study hypothesis, demonstrating that there is no relationship between the reasons for doing PA and the time spent doing it. Nonetheless, the conclusion drawn is that it is important to introduce the time variable in studies alongside the other three perspectives. "

Perhaps its the context in which this data has been analysed.  So the question here is why are people 'motivated' or not by health.  Do they have ill health, which therefore motivate them to engage in PA but not more intense or longer in time compared to another person but perhaps more from them previously in time.  motivated by health in the context of family illness, motivated in health as prevention or as treatment.  Restricted as a covariable because they have a health condition which restricts their 'time doing' PA. 

perhaps there were other confounding variables which you havent accounted for e.g. perceived competence for PA, functioning, apathy, and demographic variables?

future research consideration, what is the motivation of health on inactivity, sedentary behaviour rather than time doing PA? 

future research to explore the lack of relationship between health motivation and time doing PA, to explore participants beliefs and perceptions enabling and prohibiting PA engagement 

consider for example intrinsic/ extrinsic motivation. 

Author Response

Dear Review,

Please find enclosed the second revised version of the article, coded:  IJERPH-900946, entitled “Lack of association between the reasons for and time spent doing physical activity”, together with our response to your comments.

We are very grateful to the reviewers, whose comments have helped to improve the article. You will see that this manuscript has improved substantially. Please, we want to thank you for the time and enthusiasm you have dedicated to doing the review. The following pages describe our detailed responses to each of your comments, which have also been changed in the manuscript.

Yours sincerely,

The authors

RESPONSE TO REVIEWER’S #1 COMMENTS

Comments to the Author

Please note your review is female not male, her not his. *

Thanks for the information. We are sorry for any inconvenience we may have caused.

"Hypothesis of the study is that there are signifi... " use past tense throughout, was not is  (e.g. line 72, line 78)

We greatly appreciate your comment. The sentences have been corrected.

Line 72: The main hypothesis of the study was that there are significant differences in the time Europeans dedicate to PA depending on the motivation for doing it.

Line 78: The methodology of the study was quantitative

Line 116: amend sentence as follows please: Therefore, to the authors knowledge, there are currently no published studies at the European level tha...

We appreciate your comment and the text has been changed.

Line 116: Therefore, to the authors knowledge, there are currently no published studies at the European level that use these methodologies due to their high cost.

Line 153 "First, we carried out an explor..."  consider academic style refer reference to we.  e.g.    An exploratory factorial analysis was conducted...  or simply exploratory factorial analysis demonstrated/ revealed/ conducted .....

Thank you very much. The sentence is modified.

Line 152: An exploratory factorial analysis (Varimax rotation) was conducted to uncover the underlying structure of a reasons for doing PA. This factor structure is used in confirmatory factor analysis.

Please provide reference for Generalized Least Squares (GLS) (REF). and...
The estimation method used was the Weighted Least Squares (WLS), a method that appears in the AMOS 25 program (REF) under the label Asymptotic Free Distribution method (ADF)

Thank you very much for the comment. The reference to Generalized Least Squares (GLS) has been removed because it was an error, it has not been used in statistical analysis; and the references to the Weighted Least Squares (WLS), AMOS 25 program and Asymptotic Free Distribution method (ADF) have been added.

Line 174: The estimation method used was the Weighted Least Squares (WLS) [54], a method that appears in the AMOS 25 program [55] under the label Asymptotic Free Distribution method (ADF) [56].

These three new bibliographic references have been added:

  1. Olsson, U., Foss, T., Troye, S., & Howell, R. (2000). The Performance of ML, GLS, and WLS Estimation in Structural Equation Modeling Under Conditions of Misspecification and Nonnormality. Structural Equation Modeling-a Multidisciplinary Journal, 2000; 7, 557-595.
  2. Arbuckle, J.L. IBM SPSS AMOS 25 User’s Guide. IBM Corporation. 2017.
  3. Huang, Y. & Bentler, P.M. Behavior of Asymptotically Distribution Free Test Statistics in Covariance Versus Correlation Structure Analysis. Structural Equation Modeling: A Multidisciplinary Journal, 2015; 22:4, 489-503.

Line 264 "Of particular note are the latent variable" amend ... it is noteworthy that the latent.... 

Thank you very much. The sentences have been corrected.

Line 260: It is noteworthy that the latent variables….

Discussion: "The bibliographic references analysed showed the positive relationship between PA and health.  In other words, the people whose reason for performing PA was health should spend more time doing PA. However, the constructed model shows that this is not the case and that there is no such relationship. "

Are you suggesting here that: “Previous literature (refs from intro) suggests a positive relationship exists between PA and health.  This study hypothesised that people who are motivated by health to engage in PA, would be significantly more time doing PA.  However, this hypothesis was rejected given no such relationship was demonstrated in this sample. 

Thanks a lot for the suggestion. It has been accepted and the sentence changed.

I would really like you to expand the discussion and bring in interpretation through health behaviour theory.  For example, consider the following "out from three different perspectives. This study aimed to open up a new perspective, the study of time and motivation, and the results contribute to the literature with this new variable to consider within the study of PA. It seems logical to think that people who do PA for health reasons spend more time doing it. However, the results of the study contradict this main study hypothesis, demonstrating that there is no relationship between the reasons for doing PA and the time spent doing it. Nonetheless, the conclusion drawn is that it is important to introduce the time variable in studies alongside the other three perspectives. "

Thank you very much for your comment and for the time spent increasing the quality of the article. Suggesting his/her advice, this initial paragraph has been added:

“This study aimed to open up a new perspective, the study of time and motivation, and the results contribute to the literature with this new variable to consider within the study of PA. Specifically, the time variable in studies alongside the other three perspectives. An important limitation of the study is the lack of temporary data (at different points in time) on the spent in PA that allows to analyse the time before and after the indicated reasons appear and the previous physical experience. For example, in case of health problems: for people who indicate health reasons as a reason to perform PA, are there differences in time when compared to the time spent before having health problems?.”

Perhaps its the context in which this data has been analysed.  So the question here is why are people 'motivated' or not by health.  Do they have ill health, which therefore motivate them to engage in PA but not more intense or longer in time compared to another person but perhaps more from them previously in time.  motivated by health in the context of family illness, motivated in health as prevention or as treatment.  Restricted as a covariable because they have a health condition which restricts their 'time doing' PA. 

As said before, thank you very much for your suggestions in the Discussion section. The following paragraph has been added and the entire section has been redistributed:

“The lack of more information about the reasons. For example, in the case of the health reason, the question here is why are people 'motivated' or not by health: motivated by health in the context of family illness, motivated in health as prevention or as treatment, or have a health condition which restricts their 'time doing' PA.”

Perhaps there were other confounding variables which you haven’t accounted for e.g. perceived competence for PA, functioning, apathy, and demographic variables? Future research consideration, what is the motivation of health on inactivity, sedentary behaviour rather than time doing PA? Future research to explore the lack of relationship between health motivation and time doing PA, to explore participants beliefs and perceptions enabling and prohibiting PA engagement. Consider for example intrinsic/ extrinsic motivation. 

As said before, thank you very much for your suggestions in the Discussion section. The following paragraph has been added and the entire section has been redistributed:

“As a future line of research, it would be appropriate to analyse the motivation of health on inactivity and sedentary behaviour; and participants beliefs and perceptions enabling and prohibiting PA engagement (for example intrinsic/ extrinsic motivation).”

Reviewer 2 Report

The authors tried to edit the text according to the reviewer's comments. They have succeeded in part in this, but the discussion will still need to be adjusted. The first paragraph will require at least some citations. Similarly, I recommend moving the reflection on the limits of the study to the discussion from the conclusions and at least briefly mention the influence of age and previous physical experience on the results of the study. Of course, I would welcome a greater concretization of the results towards practical use.

Author Response

Dear Review,

Please find enclosed the second revised version of the article, coded:  IJERPH-900946, entitled “Lack of association between the reasons for and time spent doing physical activity”, together with our response to your comments.

We are very grateful to the reviewers, whose comments have helped to improve the article. You will see that this manuscript has improved substantially. Please, we want to thank you for the time and enthusiasm you have dedicated to doing the review. The following pages describe our detailed responses to each of your comments, which have also been changed in the manuscript.

Yours sincerely,

The authors

RESPONSE TO REVIEWER’S #2 COMMENTS

Comments to the Author

The authors tried to edit the text according to the reviewer's comments. They have succeeded in part in this, but the discussion will still need to be adjusted. The first paragraph will require at least some citations.

Thank you very much for your comment and for the time spent increasing the quality of the article. Suggesting his/her advice about Discussion section, the following paragraphs have been added and the entire section has been redistributed:

“This study aimed to open up a new perspective, the study of time and motivation, and the results contribute to the literature with this new variable to consider within the study of PA. Specifically, the time variable in studies alongside the other three perspectives. An important limitation of the study is the lack of temporary data (at different points in time) on the spent in PA that allows to analyse the time before and after the indicated reasons appear and the previous physical experience. For example, in case of health problems: for people who indicate health reasons as a reason to perform PA, are there differences in time when compared to the time spent before having health problems?.”

 “The lack of more information about the reasons. For example, in the case of the health reason, the question here is why are people 'motivated' or not by health: motivated by health in the context of family illness, motivated in health as prevention or as treatment, or have a health condition which restricts their 'time doing' PA.”

 “As a future line of research, it would be appropriate to analyse the motivation of health on inactivity and sedentary behaviour; and participants beliefs and perceptions enabling and prohibiting PA engagement (for example intrinsic/ extrinsic motivation).”

The Discussion section has been rewritten in this way:

This study aimed to open up a new perspective, the study of time and motivation, and the results contribute to the literature with this new variable to consider within the study of PA. Specifically, the time variable in studies alongside the other three perspectives. An important limitation of the study is the lack of temporary data (at different points in time) on the spent in PA that allows to analyse the time before and after the indicated reasons appear and the previous physical experience. For example, in case of health problems: for people who indicate health reasons as a reason to perform PA, are there differences in time when compared to the time spent before having health problems?.

The first objective set in this study was to develop a factorial structure that reduced the number of reasons why Europeans do PA. The results showed a robust factorial structure made up of 6 dimensions: fitness and health, appearance, fun and friend (F&F), self-esteem, aging, and relaxation. This dimensional structure coincided with the motivational structures studied in the bibliography review [22], allowing us to create a factorial model that summarised the reasons for doing PA.

The second objective was to determine if there were two different model structures for vigorous PA and moderate PA, and if the model was sound, or in other words, to identify if the motivational structure was different for people doing vigorous PA and moderate PA. As the confirmatory analysis showed, there were not two different structures and the two types of activity were explained by the same measurement model. The measurement model was robust and explained this behaviour (the correlation between the times implied that both vigorous and moderate exercise were explained by the same model).

Dimension F2, physical appearance, was highly correlated with the dimension of health and exercise. Therefore, the Europeans that stated that they do PA for their appearance and to indirectly acquire new skills were also doing it for health reasons. The model also showed another correlation between the factors of appearance (F2) and self-esteem (F5). This result was to be expected since people concerned with their appearance and who do PA to improve it are also indirectly improving their self-esteem. These correlations are important given that they can be used to re-focus policies to incentivise doing PA to emphasise not only the health aspect but also the factors of appearance and self-esteem.  

The third and last objective was to analyse whether the reasons for doing PA explained the time spent doing it, either vigorously or moderately. Previous literature suggests a positive relationship exists between PA and health.  This study hypothesised that people who are motivated by health to engage in PA, would be significantly more time doing PA.  However, this hypothesis was rejected given no such relationship was demonstrated in this sample. The lack of more information about the reasons. For example, in the case of the health reason, the question here is why are people 'motivated' or not by health: motivated by health in the context of family illness, motivated in health as prevention or as treatment, or have a health condition which restricts their 'time doing' PA.

The last structural equations model indicated that there was no relationship between them, or in other words that the analysis of the reasons for doing PA did not involve studying the time spent doing it. This finding suggests that to analyse the time spent doing PA, it would be more appropriate to relate it to other types of variables such as demographic ones or types of diseases, among others. Incorporating the time spent doing PA into studies on PA is also a new line of investigation that can help to know the phenomenon better and can better incentivise the public policies that promote doing PA, thus improving the health of the population. The time spent doing PA and the times of day must be known to create both public and private spaces that help to promote PA and, indirectly, people’s health, social integration, and self-esteem.

The main limitation of the study is the way the survey was carried out, to the effect that when asked their reasons for doing PA respondents were not asked to differentiate between vigorous and moderate PA. Therefore, their self-reported reasons were for doing PA in general. Other limitations stem from the construction of the variable “time spent”, and from the subsample chosen, which was made up of Europeans that do PA and who had done vigorous or moderate PA in the previous seven days.

As a future line of research, it would be appropriate to analyse the motivation of health on inactivity and sedentary behaviour; and participants beliefs and perceptions enabling and prohibiting PA engagement (for example intrinsic/ extrinsic motivation).

Similarly, I recommend moving the reflection on the limits of the study to the discussion from the conclusions and at least briefly mention the influence of age and previous physical experience on the results of the study. Of course, I would welcome a greater concretization of the results towards practical use.

Thank you very much for your comments. The quality of the article has been followed and increased.
